# SMOOTH MATHEMATICAL FUNCTION FROM COMPACT NEURAL NETWORKS

## ABSTRACT

This is paper for the smooth function approximation by neural networks (NN). Mathematical or physical functions can be replaced by NN models through regression. In this study, we get NNs that generate highly accurate and highly smooth function, which only comprised of a few weight parameters, through discussing a few topics about regression. First, we reinterpret inside of NNs for regression; consequently, we propose a new activation function–integrated sigmoid linear unit (ISLU). Then special charateristics of metadata for regression, which is different from other data like image or sound, is discussed for improving the performance of neural networks. Finally, the one of a simple hierarchical NN that generate models substituting mathematical function is presented, and the new batch concept "meta-batch" which improves the performance of NN several times more is introduced. The new activation function, meta-batch method, features of numerical data, meta-augmentation with metaparameters, and a structure of NN generating a compact multi-layer perceptron(MLP) are essential in this study.

## 1 INTRODUCTION

In many fields, such as astronomy, physics, and economics, someone may want to obtain a general function that satisfies a dataset through regression from numerical data, which are fairly accurate (Ferrari & Stengel (2005); Czarnecki et al. (2017); Raissi et al. (2019); Langer (2021)). The problem of smoothly approximating and inferring general functions using neural networks (NNs) has been considered in the some literature. However, there is insufficient research on using NNs to completely replace the ideal mathematical functions of highly smooth levels, which are sufficiently precise to be problem-free when a simulation is performed. This study aims to completely replace such ideal mathematical functions.

Assuming a model $M(X)$ was developed by regression on a dataset using an NN. $M(X)$ for input $X$ can be thought of as a replacement of a mathematical function $f(X)$. In this study, such NN is called "*neural function(NF)*" as a mathematical function created by an NN. The components of an analytic mathematical function can be analyzed using a series expansion or other methods, whereas it is difficult for a NF.

In this study, we *created "highly accurate" and "highly smooth" NFs with a "few parameters" using metadata.* Particularly, we combined *a new activation function, a meta-batch method, and weight-generating network (WGN)* to realize the desired performances.

The major contributions of this study can be summarized as follows.

- We dissected and interpreted the middle layers of NNs. The outputs of each layer are considered basis functions for the next layer; from this interpretation, we proposed a *new activation function*–integrated sigmoid linear unit (ISLU)-suitable for regression.

- The characteristics and advantages of metadata for regression problems were investigated. A training technique with *fictious metaparameters and data augmentation*, which significantly improves performance, was introduced. It was also shown that for regression problems, *the function values at specific locations* could be used as metaparameters representing the characteristics of a task.

- NN structures that could *generate compact*[1] *NFs* for each task from metaparameters were investigated, and a new batch concept– *'meta-batch'*–that could be used in the NFs was introduced.

## 2 NNs FOR REGRESSION

Let's talk about an easy but non-common interpretation about regression with a multi-layer perceptron (MLP). What do the outputs of each layer of an MLP mean? They can be seen as *basis functions that determine the function to be input to the next layer*. The input $x_{i+1}$ of the $(i+1)$th layer can be expressed as follows:

$$x_j^{i+1} = \sum_k w_{j,k}^i * M_k^i(x_0) + b_j, \qquad (1)$$

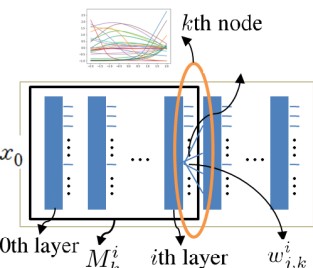

where $x_0$ denotes the input of the first layer, $w_{j,k}^i$ denotes the weight that connects the $k$th node of the $i$th layer to $j$th node of the $(i+1)$th layer, and $M_k^i$ denotes a model comprising the 0th to $i$th layers and having the $k$th node of the $i$th layer as the output. This is similar to the expression $f(x) = \sum_j w_j \phi_j(x) + b$ of the radial basis function(RBF) kernel method. Clearly, the outputs of each layer act as basis functions for the next layer. Figure 2 shows the outputs of each layer of an MLP that learned the dataset $D = \{(x_i, y_i)|y = 0.2(x-1)x(x+1.5), x \in [-2, 2]\}$ with the exponential linear unit (ELU) activation function.

Figure 1: Perspective on MLP

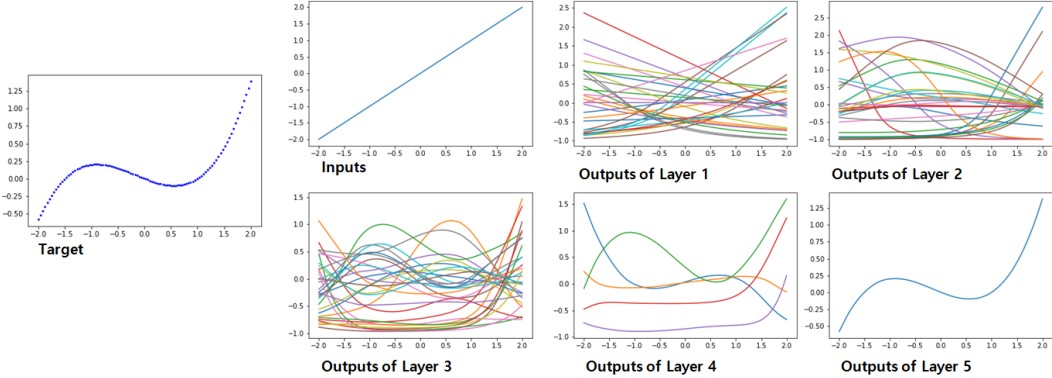

Figure 2: The output graphs of each layer, trained with an MLP, where the nodes of each layer are [1,30,30,30,5,1].

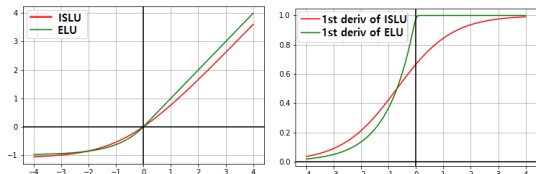

Figure 3: The graphs of ELU and ISLU($\alpha = 0.5$, $\beta = 1$)

To efficiently extract the final function, the output functions of the intermediate layers must be well-developed. If the output functions of each layer are well-developed, the desired final NF can be compact. In addition, for the final function of NN to be infinitely differentiable, the output functions of the intermediate layers should also be infinitely differentiable.

If the activation function is a rectified linear unit(ReLU), the output function bends sharply after every layer. If a one-dimensional regression problem is modeled with a simple MLP that has (k+1) layers with nodes $[N_0, N_1, N_2..N_k]$, the output function will bend more than $N_0 * N_1...N_k$. The ELU activation function weakens such bending but does not smoothen it for the

---

[1]Comprising few parameters

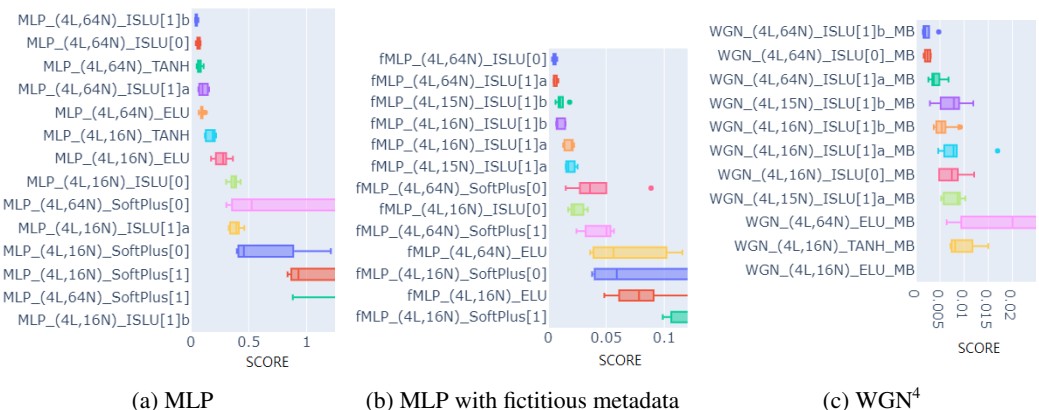

(a) MLP      (b) MLP with fictitious metadata      (c) WGN[4]

Figure 4: Scores. The numerical score table is shown in Appendix E.

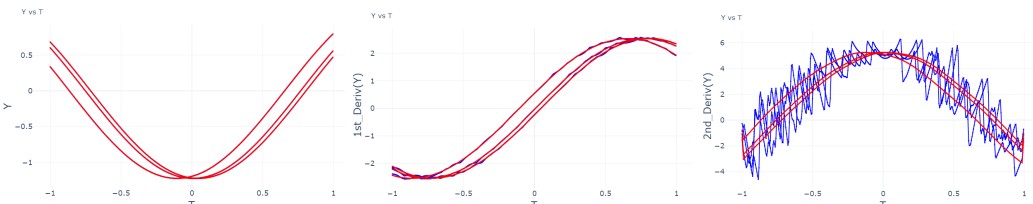

Figure 5: Comparison of ELU and ISLU when training with WGN. From left to right, the 0th, 1st, and 2nd derivatives of the curves with respect to time t in a task in the given metadatasets. Blue lines: WGN_(4L,64N)_ELU_MB, Red lines: WGN_(4L,64N)_ISLU[1]a_MB

first derivative. Moreover, apt attention is required when using the hyperbolic tangent function for all layers in a regression problem because the output function bends in two places after each layer.

Thus, the question is which activation function can develop the intermediate basis functions well? If the activation function starts as a linear function and bends at an appropriate curvature after each layer, the final result will be good. Therefore, we propose an activation function suitable for regression, called *"integrated sigmoid linear unit(ISLU)"*.

$$\log(\alpha + exp(\beta x))/\beta - \log(1 + \alpha)/\beta, \qquad (2)$$

where $\alpha$ and $\beta$ are positive numbers.

Our experiment shows that ISLU performs sufficiently well and is worth further research. It can improve the accuracy and smoothness of our experimental data. [2] Mathematically, ISLU for $\alpha = 1$ is a translated SoftPlus that passes the origin, but ISLU absolutely differs from SoftPlus. The purposes of their production differ, and there is a significant difference in their results.[3]

The experimental results are shown in Figure 4.[5] [6] [7] By default, a model structure is represented in the form "[the name of the model structure]_([the number of layers]L, [the number of nodes of all hidden layers)N]_[activation function]_[further information (option)]." The experimental metadataset is described in Appendix A <1>, which has $B$, $k$, and $m$ as metaparameters and the corresponding task

---

[2]Numerical score discussion for smoothness is presented in Appendix F.

[3]A detailed explanation of this is presented in Appendix C.

[4]With SoftPlus[1], WGN could not be trained due to the divergence of loss.

[5]In our experiment, the Swish activation function was also tested, and its performance was comparable to that of ISLU. However, for consistency, we do not discuss it in the main text; the details are presented in Appendix B.

[6]All box plots in this study are arranged in order of small scores from the top, and items wherein the box is invisible have a larger score than the shown graph range.

[7]All experimental conditions of NNs in this study are shown in Appendix D

dataset for $L,t,\phi$. [8] The average of the "sum of error squares" for eight tasks among the experimental metadatasets is considered a score.

In Figure 4a, we consider a basic MLP structure trained on one task; WGN and fMLP, which will be introduced hereinafter, in Figure 4b,4c are models trained using metadata. Considering ISLU[0], what is in [] represents a degree of freedom in which the activation function's shape can be changed. ISLU[0] is trained with $\alpha = 0.5$ and $\beta = 1$, ISLU[1]$_a$ is trained with $\alpha = 0.5$ and $\beta = var$, and ISLU[1]$_b$ is trained with $\alpha = 0.5$ and $\beta = 1 + var$, where $var$ are trainable parameters. Because variables tend to be learned in a distribution with a mean near zero when training with an NN, ISLU[1]$_a$ bends slightly after each layer and ISLU[1]$_b$ bends to a certain amount and additionally adjusts its degree. [9]

Considering the experimental results in Figure 4, the following is observed.

- (1) There is a significant difference in performance between SoftPlus and ISLU.
- (2) Considering an MLP, there is not much difference in performance between ISLU and ELU (Figure 4a). However, in all models trained with metadata, ISLU significantly outperforms ELU (Figure 4b,4c).
- (3) In Figure 4b, when the number of nodes is high(64N), ISLU[0] outperforms ISLU[1], whereas when the number of nodes is low(15N,16N), ISLU[1] outperforms ISLU[0].
- (4) In Figure 4c, ISLU[1]$_b$ always outperforms ISLU[0].
- (5) As shown in ISLU[1]$_a$ and ISLU[1]$_b$, there are slight differences in performance depending on what the shape of ISLU is based on.

The reason for (2) can be explained as follows: setting an activation function parameter entails giving a certain bias. When given well, it considerably helps in predicting results; otherwise, it may interfere. When using metadata, the performance is improved because biases are determined by referring to various data.

We now discuss the reasons for (3) and (4). In Figure 4b, fMLP indicates an MLP structure trained with fictitious metadata[10] for only one task. If an MLP has a lots of nodes, even if the curvature functions of all activations are set to be the same, several functions can be added and combined to produce curves with the desired shapes. Meanwhile, when the nodes are few, desired curves may not be obtained well without adjusting the curvatures of the activation functions. In Figure 4c, WGN is a network structure [11] that learns the entire metadata at once. In this case, using ISLU[1] allows the activation shape to change between tasks, yielding better results than the fixed-shaped ISLU[0].

The ISLU presented in this study is an example of an activation function for creating desired curves; a better activation function can be studied.

## 2.1 PERSPECTIVES OF METADATA

In this study, *metadata* are the data of datasets that are sometimes the sets of task datasets, *metafeatures* are features of a task dataset, and *metalabels* or *metaparameters* are parameters representing metafeatures. Consider a case where a physical system has the relation $y = f(x_1, x_2..)$ and the function $f$ depends on the variables $a1, a2...$. For example, a pendulum's kinetic energy $E$ is $E = f(\theta)$, where $\theta$ denotes the angle between the string and gravitational field direction, and the function $f$ depends on the string's length $l$ or pendulum's mass $m$.

In this case, the kinetic energy $E$ can be viewed not only as $f(\theta, l, m..)$ but also as $f_{l,m}(\theta)$. The dataset $\mathcal{D} =$

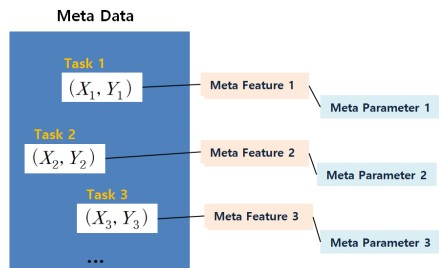

Figure 6: Metadata structure.

---

[8]Most of the experiments in this study are done with this experimental dataset.

[9]The smaller the $\beta$ value, the closer ISLU is to a straight line.

[10]described in 2.2

[11]described in 3.1

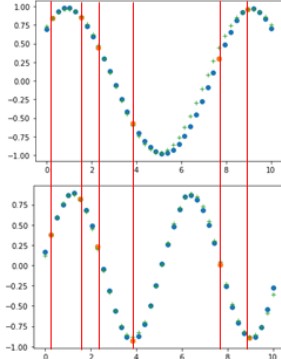

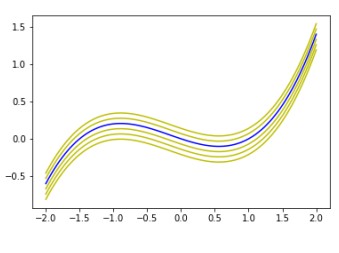

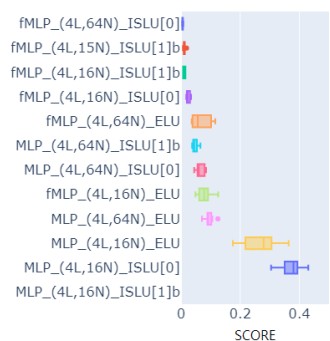

Figure 7: The NN learns the sine curves even if all the task dataset's x values for any $A, p, \phi$ are only at x=0.26, 1.54, 2.30, 3.84, 7.69, 8.97.

Figure 8: Meta-augmentations *with* fictitious metalabels. The meta-augmentations of the y-axis are displayed.

Figure 9: Performance improvement when using fictitious metaparameters. "fMLP" indicates MLP trained with using fictitious metaparameters.

$\{(l_i, m_i, \theta_i, E_i)|E_i = f(\theta_i, l_i, m_i..)\} = \{(l_i, m_i, D_i)|D_i = D_{m_i, l_i}(\theta)\}$ is metadataset and the numerical value $l, m$ can be considered as metaparameters.

One might want to interpret the kinetic energy as $E = f_{l,\theta}(m)$. This cannot be said to be wrong, and there may be various perspectives and interpretations for *a numerical dataset used for regression*.

## 2.2 ADVANTAGES OF TRAINING WITH METADATA AND META-AUGMENTATION

Consider an experiment performed with the following metadata $\mathcal{D}_k = \{(x_i, y_i)|y_i = A_k * \sin(p_k * x_i + \phi_k), x \in [0, 10], A_k \in [-1.5, 1.5], p_k \in [0.5, 1.5], \phi_k \in [0, 2\pi]\}$. It can be seen from the perspective that the tasks $\mathcal{D} = \{(x_i, y_i)|y_i = A * \sin(p * x_i + \phi)\}$ are given according to the metaparameters of $A$, $p$, and $\phi$. In this case, if not only $x$ but also $A$, $p$, and $\phi$ were trained as training inputs, a curve could be created with zero shot just by setting $A$, $p$, and $\phi$. [12] Consequently, if metadata are used to learn, the accuracy of each task increases.

Taking a hint from the fact that metadata improve inference accuracy for each task, it can be thought that even in a situation where only one task is given, fictitious metadata with fictitious metalabels (or metaparameters) can be generated to learn curves. If only fictitious metalabels are used and the data remain the same, curves would be learned in the direction of ignoring the metalabels; therefore, some data modifications are required. For the experiment, fictitious metadata comprising 10 tasks with the metaparameter $a$ were created by moving the $y_i$ value in parallel $\pm 0.05$ for every $a = \pm 0.02$ with the original data of $a = 0$ for a given task $\mathcal{D} = \{(x_i, y_i)\}$. As a result of using fictitious metadata, the score improved significantly (Figure 9). The performance improvement was similar even when the fictitious metadata were generated by moving $x_i$ instead of $y_i$ according to the fictitious metalabel.

We reiterate that data augmentation *including ficitous meta-parameters* is required to achieve significant performance improvement, otherwise there is little performance improvement. In this study, only the experimental results using MLP with fictitious metaparameters added to inputs are shown; however, further experiments show that the performance improvement due to fictitious metadata occurs independent of the model structure.

## 2.3 LEARNING FUNCTION WITH RESTRICTED METADATA

The regression task for the numerical dataset $\mathcal{D} = \{(x_i, y_i)|i = 0, 1, 2..\}$ can have a significant advantage different from the image problems, i.e., $y_i$ *values at particular locations can be metaparameters that represent the entire task dataset*. For the set of images, *if we know the RGB values at specific positions of pixels, it does not help to distinguish the features of images*. However, for a set of mathematical functions f(x)s such as fifth degree polynomial or sine curve sets, *just knowing f(x) at*

---

[12]MLP with inputs $A, p, \phi, \theta$ and the WGN in 3.1 were used for the experiment.

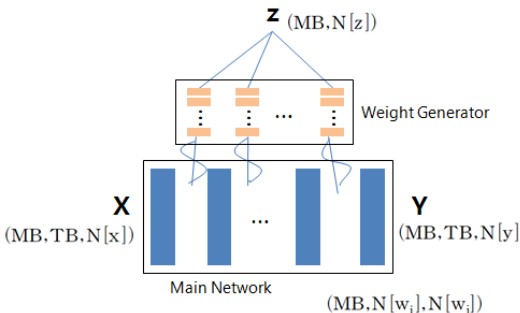

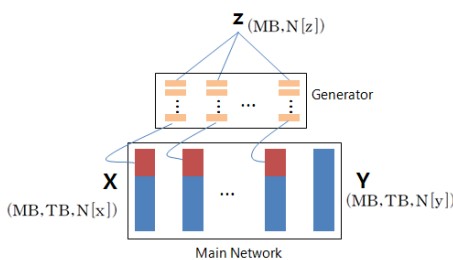

Figure 10: WGN structure and training with meta-batch, where z denotes a metaparameter, X denotes the input, Y denotes the output, and w denotes the weight of main Network.

Figure 11: Another example of function-generating networks where meta-batch can be used.

*specific x positions can let us distinguish the functions well*. This can be shown in the experiments with sine curve datasets. For the tasks $\mathcal{D}_k = \{(x_i, y_i)|y_i = A_k * \sin(p_k * x_i + \phi_k), x \in [0, 10], A_k \in [-1.5, 1.5], p_k \in [0.5, 1.5], \phi_k \in [0, 2\pi]\}$ that are not given metaparameters $A$, $p$, and $\phi$, it is possible to learn the sine curves just using the function values $y_i$ at six points of $x_i$ as metaparameters (Figure 7). In other words, it is possible to perform *few-shot learning* simply without metalabels.

In addition, the relationship between the six $y$ points and $A$, $p$, and $\phi$ can be learned with a simple MLP that has six-dimensional inputs and three-dimensional outputs, indicating that the metaparameters $A$, $p$, and $\phi$ can be completely extracted to generate a sine curve using on the six points.

## 3 FUNCTION-GENERATING NETWORKS

### 3.1 WGN

When learning metadata in a regression problem, one can think of an hierarchical NN structure in which a NF corresponding to each task is generated from corresponding meta parameters. The structure in which a model is generated from variables has been studied extensively (Rusu et al. (2018); Sun et al. (2019)). We consider the one of the structure of a function-generating network called *weight generating network(WGN)* in this study. As shown in Figure 6, WGN generates parameters such as the weight and bias of *main network* through a simple MLP called *weight generator* from metaparameters. If there are trainable parameters of the activation function on the main network, can also be generated from metaparameters.

WGN is expected to generate *NFs comprising a few parameters* corresponding to each task through the weight generator.This is because enormous data and weight generators carefully generate the parameters of the main network. Experiments showed that WGN is effective in creating the main network with excellent performance, although it comprises only a few parameters.

What are the advantages of creating a NF with *only a few parameters*? First, because the number of times that a linear input function can be bent is reduced, it may have a regulation effect or help create a smooth function. Second, it may be helpful in interpreting and analyzing the network by directly adjusting the parameters. Third, because the number of weights is small and the inference speed is fast, it can be advantageous when a fast interference speed is required, such as a simulation.

### 3.2 META-BATCH

When training a function-generating network, such as WGN, 'one' metalabel (or metaparameter) $z_i$ is usually placed on the weight generator's input, and it is updated with the batch of the corresponding task on the main network. However, in this case, it becomes training with batch-size=1 for the metaparameters, and when it is updated with backpropagation at once, the metacorrelation between tasks is not used well. From these problems, the *meta-batch* concept is proposed. To distinguish the

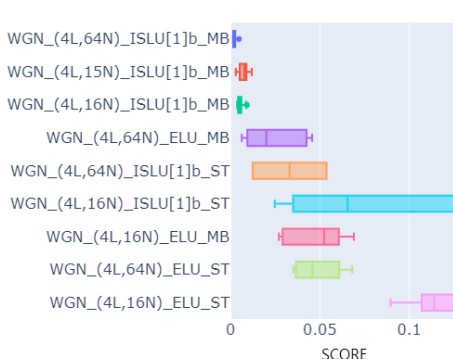

Figure 12: Comparison between using meta-batch and not using meta-batch

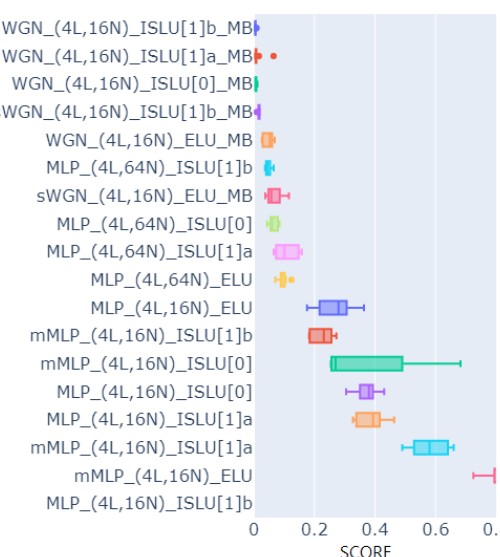

Figure 13: Scores for each task of metadata from different models.

*meta-batch* from the conventional batch, the batch of each task corresponding to one $z_i$ is called "task batch." "Meta-batch" refers to both the batch of metaparameters and the corresponding batch of the tasks. The training method for WGN using the meta-batch is as follows.

Suppose a training metadataset $\mathcal{D} = \{(\mathcal{D}_k, z_k) | k \in \{1..K\}\}$ comprising task training datasets $\mathcal{D}_k = \{(x_i^k, y_i^k)\}_{i=1}^{N_k}$ are given, where $N_k$ is the number of datapoints of $\mathcal{D}_k$ task. For index sets $M \subset \{1,,,K\}, T_k \subset \{1,,,N_k\}$ that determines meta-batch and task batch, select the batch $\mathcal{X}_M = \{(D_m, z_m) | m \in M\}$ and $\mathcal{X}_T^M = \{(x_t^m, y_t^m) | t \in T_m, m \in M\}$.

We denote the dimensions of $x_i, y_i$, and $z_i$ as $N[x], N[y]$, and $N[z]$, respectively. $w_{ij}^l$ denotes the weight between the $l$th and $(l + 1)$th layers of the WGN's main network, which has a shape $(N[w_l], N[w_{l+1}])$, where $N[w_i]$ denotes the number of nodes at the $i$-th layer. The inputs $\mathcal{X}_T^M$ of the main network are rank-3 tensors in the form of $(\text{MB}, \text{TB}, N[x])$, where MB and TB denote the sizes of $M$ and $T$, respectively.

If $z_m$ enters to weight generator as inputs in the form of $(\text{MB}, N[z])$, $G[w_{ij}^l](z_m)$ generates a tensor in the form $(\text{MB}, N[w_l] * N[w_{l+1}])$ and it is reshaped as $(\text{MB}, N[w_l], N[w_{l+1}])$, where $G[w_{ij}^l]$ denotes a generator that generates $w_{ij}^l$. The outputs of the $l$-th layer of the main network, which has the shape $(\text{MB}, \text{TB}, N[w_l])$, are matrix-producted with the weights in the form $(\text{MB}, N[w_l], N[w_{l+1}])$, and then it becomes a tensor in the form $(\text{MB}, \text{TB}, N[w_{l+1}])$.[13] Finally, the outputs of the main network with shape $(\text{MB}, \text{TB}, N[y])$ and $y_t^m$ are used to calculate the loss of the entire network. Conceptually, it is simple as shown in Figure 10.

As a result of the experiment, Figure 12 [14] shows a significant difference in performance between using and not using meta-batch, where "MB" means using meta-batch, and "ST" means training by inputting metaparameters individually without using meta-batch. Figure 12 also shows the difference between using WGN and just using a simple MLP.

Meta-batch can be used in any function-generating network structure that generates models from variables; another example is shown in Figure 11. The outputs of generators concatenate with the layers of the main network. As a result of experimenting with ISLU[1] in the structure shown in Figure 11, there was a performance difference of more than four times between using and not using meta-batch.

---

[13]All other parameters in the main network can be generated from weigh generators using a similar method

[14]For a WGN, an MLP with three hidden layers and 40 nodes was used as the weight generator.

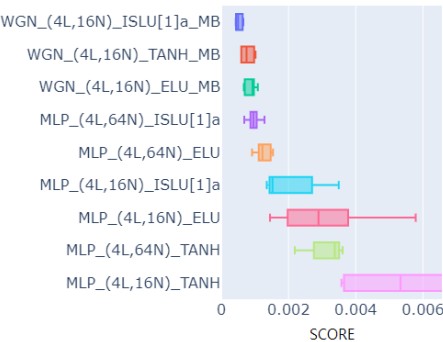

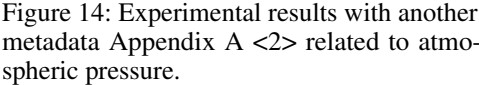

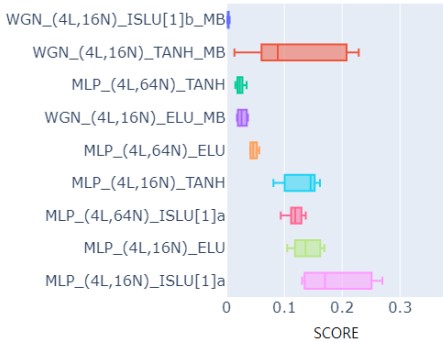

Figure 14: Experimental results with another metadata Appendix A <2> related to atmospheric pressure.

Figure 15: Experimental results with metadata Appendix A <3> comprising sine and cosine functions

Figure 13 shows the results of using WGN and meta-batch compared with those of using only MLP. "sWGN" indicates a WGN trained with metaparameters that are the function values at 10 points of $(L, t, \phi)$ without using original metaparameters "$B$, $k$, and $m$." "mMLP" indicates an MLP that trained with a six-dimensional input combined with "$L$, $t$, and $\phi$" and the original metaparameters. "MLP" indicates a trained model for each task with just inputs "$L$, $t$, and $\phi$." This figure shows that using meta-batch, WGN outperformed MLP with fewer parameters. This also shows that WGN excels at learning all metadata and using them with only a few parameters.

Figures 14 and 15 shows the results of other metadatasets, which are described in Appendix A. The combinations of ISLU, meta-batch, and WGN give much better performance than MLP in terms of accuracy and compactness.

## 4 CONCLUSION

In this study, we focus on creating mathematical functions with desired shapes using an NN with a few parameters. Irregular and numerous parameters are helpful for generalizations because of randomness; however, this sometimes makes it difficult to interpret the network and reduces the smoothness of the functions.

In this study, we dissected NNs for regression; consequently, we proposed a new activation function. We looked at the special features of regression-related metadata, such as the possibilities to extract meta-parameters immediately, and how, given only one task, we could create ficitious meta-parameters and metadata to increase performance by more than a few times.

In addition, the network structures generating NFs from metaparameters were discussed and the *meta-batch* method was introduced and tested for the structure called WGN. WGN makes it possible to provide smooth and desired-shaped NFs comprised of a few parameters because it carefully generates different parameters and shapes of activation functions for each task.

The findings of this study, as well as the insights obtained in the process, are significant for earning smooth and accurate functions from NNs. One of them is the perspective of obtaining desired output functions at *intermediate* layers from enormous data. Regarding regression problems, it will help elucidate how to find the metafeature of each task and map to the corresponding metaparameter as well as how to get a smooth and compact NF of a desired shape.

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

# APPENDIX

## A    THE EXPERIMENTAL METADATASET

<1> The following dataset was generated from the formula in the physics book of a graduate school.[15] The original problem and its answer are as follows.

---

"A spring is connected to a support at one end and has a mass $m$ attached at the other, where the spring constant is $k$ and the rest length is $L$. Neglecting the spring's mass, what is the angular position $\theta$ of mass $m$ under the gravitational field as a function of time $t$?"

=> **answer** : $\theta = B \cos{(\sqrt{\frac{kg}{kL+km}}t + \phi)}$, where $B, \phi$ are constants of integration.

---

The formula $\theta = B \cos{(\sqrt{\frac{kg}{kL+km}}t + \phi)}$ was slightly modified, and the following dataset is generated.

$$\mathcal{D} = \{(B_i, k_i, m_i, L_i, t_i, \phi_i, \theta_i) | \theta_i = \sqrt{B_i} \cos{(\sqrt{\frac{k_i g + (B_i - 0.3)^2}{k_i L_i + k_i m_i}}t_i + \phi_i)}, B_i \in [0.5, 1.5],$$

$$k_i \in [2, 5], g = 9.8, m_i \in [0.5, 2.5], L_i \in [1, 4], t_i \in [0.1, 2], \phi_i \in [0, 0.78]\}$$

$$\sim \{(B_i, k_i, m_i, \mathcal{D}_{B_i, k_i, m_i})\}$$

When it was trained, all input variables $B, k, m, L, t, \phi$ were normalized.

For each of $B$, $k$, and $m$, 10 points were uniformly selected to make 1,000 metaparameter sets $\{(B_i, k_i, m_i)\}$, and for each metaparameter point, task datasets $\mathcal{D}_{B_i, k_i, m_i} = \{(L_i, t_i, \phi_i, \theta_i)\}$ which have 35,301 points were created by selecting 21, 41, and 41 uniform points of $L, t$, and $\phi$, respectively. Among them, 100 random metaparameters were selected, and 640 points were selected for each task to be used as training metadata. The selected points are shown in Figure 16.

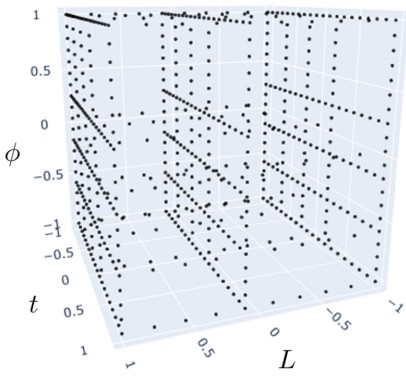

Figure 16: Points of training datasets.

<2> The dataset for Figure 14 is

$$\mathcal{D} = \{(B_i, k_i, m_i, L_i, t_i, \phi_i, y_i) | y_i = \frac{-m_i g}{B_i k_i} \log{\frac{t_i}{t_i - B_i L_i}} + \log{\phi_i}, B_i \in [1, 10],$$

$$k_i \in [5, 20], g = 9.8, m_i \in [1, 20], L_i \in [1, 8], t_i \in [100, 150], \phi_i \in [2, 20]\}$$

$$\sim \{(B_i, k_i, m_i, \mathcal{D}_{B_i, k_i, m_i})\}$$

---

[15]Goldstein, H., Poole, C., & Safko, J. (2002). Classical mechanics.

<3> The dataset for Figure 15 is

$$\mathcal{D} = \{(B_i, k_i, m_i, L_i, t_i, \phi_i, y_i) | y_i = B_i \sin(k_i t_i + B_i) + m_i \cos^2(\phi_i L_i + m_i), B_i \in [-0.4, 0.4],$$
$$k_i \in [1, 1.5], g = 9.8, m_i \in [0.2, 1], L_i \in [0, 1.4], t_i \in [0, 1.4], \phi_i \in [1, 1.5]\}$$
$$\sim \{(B_i, k_i, m_i, \mathcal{D}_{B_i, k_i, m_i})\}$$

## B  ABOUT SWISH ACTIVATION FUNCTION

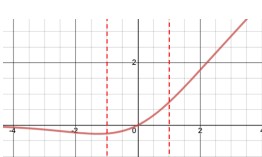

Figure 17: Swish.

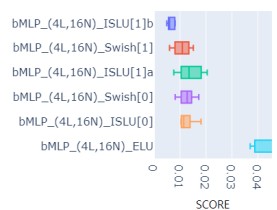

Figure 18: Scores of bMLP with the amplified data.

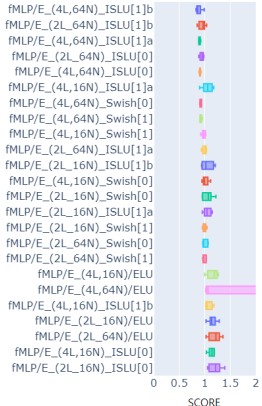

Figure 19: Scores of fMLP with errored data.

Swish showed similar or sometimes slightly better performance than ISLU in our given experimental data. Perhaps the reason is that Swish is good for generalization, and our data fall within the smooth range of Swish activation $(-1, 1)$.

If the data targets are in the range of $(-1, 1)$, Swish may have to be used in a range that includes two bends (inflection points), which may result in a slightly worse performance. In actual experiments, Swish underperformed ISLU when the experimental data targets had values significantly outside the range of $(-1, 1)$. Figure 18 shows the experimental results of data obtained by increasing the target value by 20 times that of the data in Appendix A <1>, where bMLP means fMLP [16] with the new data whose targets are amplified.

In addition, if the data are mixed with some errors, ISLU tends to slightly outperform Swish. Figure 19 shows the scores when training fMLP with the experimental data obtained by adding random errors to the original data targets within 1%.[17] This shows ISLU slightly outperforms Swish, possibly because ISLU's curve is simpler and gives a regularization effect.

## C  REASON FOR PERFORMANCE DIFFERENCE BETWEEN ISLU AND SOFTPLUS

Even when comparing ISLU[0] with $\alpha = 1$ and $\beta = 1$ and SoftPlus, there are differences in performance because the parameters try to follow a specific distribution when an NN is trained.

Particularly, for two activation functions $AF$ and $AF' = AF + b$, when the parameter $w$ is multiplied, there is a difference in parallel movement by only $w * b$ because $w * AF'(x) = w * (AF(x) + b) = w * AF(x) + w * b$. It may be thought that the bias parameter could be adjusted to produce the same performance; however, there are differences in performance because the NN parameters prefer a certain distribution. Further, because $ISLU(x, \beta) = SoftPlus(x, \beta) + b(\beta)$, the differences widen because $\beta$ is entangled with the translation part.

---

[16]MLP with meta-augmentation in 2.2

[17]Data mixed with random errors were generated only once and applied in all cases.

# D  TRAINING CONDITIONS

The model with the structure denoted by (pL,qN) means the model where the number of hidden layers, excluding the input and output layers, is p-1, and the total number of layers is p+1.

For WGN, all weight generators were configured separately for weight, bias, and activation function parameters of the main networks, all of which were MLPs with 40 nodes and 2 hidden layers. The activation function of the generator did not significantly affect the results of either ISLU[0], ISLU[1], or Swish; all of them were set to Swish for consistency. The meta-batch size was 16, and the running rate was 0.88 times for every 5,000 updates starting from 0.001, with a total of 350,000 updates.

For the model structure in Figure 11, all settings were the same as above except that the generators only generate the inputs of each layer, which are concatenated with the other inputs of each layers of the main network. Each eight-dimensional input of the main network was generated by the generators, and experiments were performed with the (4L,16N) structure.

# E  SCORES FOR FIGURES

Table 1: Table for Figure 4a

| Model | Score |
|---|---|
| MLP_(4L,64N)_ISLU[1]b | 0.0480 |
| MLP_(4L,64N)_ISLU[0] | 0.0650 |
| MLP_(4L,64N)_TANH | 0.0746 |
| MLP_(4L,64N)_ELU | 0.0949 |
| MLP_(4L,64N)_ISLU[1]a | 0.1087 |
| MLP_(4L,16N)_TANH | 0.1735 |
| MLP_(4L,16N)_ELU | 0.2674 |
| MLP_(4L,16N)_ISLU[0] | 0.3722 |
| MLP_(4L,16N)_ISLU[1]a | 0.3853 |
| MLP_(4L,16N)_SoftPlus[0] | 0.6513 |
| MLP_(4L,16N)_SoftPlus[1] | 1.5230 |
| MLP_(4L,16N)_ISLU[1]b | 14799.9304 |
| MLP_(4L,64N)_SoftPlus[0] | 314.3878 |
| MLP_(4L,64N)_SoftPlus[1] | 329.3784 |

Table 2: Table for Figure 4b

| Model | Score |
|---|---|
| fMLP_(4L,64N)_ISLU[0] | 0.0055 |
| fMLP_(4L,64N)_ISLU[1]a | 0.0062 |
| fMLP_(4L,16N)_ISLU[1]b | 0.0103 |
| fMLP_(4L,15N)_ISLU[1]b | 0.0112 |
| fMLP_(4L,16N)_ISLU[1]a | 0.0181 |
| fMLP_(4L,15N)_ISLU[1]a | 0.0191 |
| fMLP_(4L,16N)_ISLU[0] | 0.0247 |
| fMLP_(4L,64N)_SoftPlus[0] | 0.0418 |
| fMLP_(4L,64N)_SoftPlus[1] | 0.0438 |
| fMLP_(4L,64N)_ELU | 0.0691 |
| fMLP_(4L,16N)_ELU | 0.0793 |
| fMLP_(4L,16N)_SoftPlus[1] | 0.1306 |
| fMLP_(4L,16N)_SoftPlus[0] | 230.7397 |

Table 3: Table for Figure 4c

| Model | Score |
|---|---|
| WGN_(4L,64N)_ISLU[1]b_MB | 0.0024 |
| WGN_(4L,64N)_ISLU[0]_MB | 0.0024 |
| WGN_(4L,64N)_ISLU[1]a_MB | 0.0042 |
| WGN_(4L,16N)_ISLU[1]b_MB | 0.0058 |
| WGN_(4L,15N)_ISLU[1]b_MB | 0.0073 |
| WGN_(4L,16N)_ISLU[0]_MB | 0.0074 |
| WGN_(4L,15N)_ISLU[1]a_MB | 0.0078 |
| WGN_(4L,16N)_TANH_MB | 0.0097 |
| WGN_(4L,16N)_ISLU[1]a_MB | 0.0138 |
| WGN_(4L,64N)_ELU_MB | 0.0248 |
| WGN_(4L,16N)_ELU_MB | 0.0473 |

Table 4: Table for Figure 9

| Model | Score |
|---|---|
| fMLP_(4L,64N)_ISLU[0] | 0.0055 |
| fMLP_(4L,16N)_ISLU[1]b | 0.0103 |
| fMLP_(4L,15N)_ISLU[1]b | 0.0112 |
| fMLP_(4L,16N)_ISLU[0] | 0.0247 |
| MLP_(4L,64N)_ISLU[1]b | 0.0479 |
| MLP_(4L,64N)_ISLU[0] | 0.0650 |
| fMLP_(4L,64N)_ELU | 0.0691 |
| fMLP_(4L,16N)_ELU | 0.0793 |
| MLP_(4L,64N)_ELU | 0.0949 |
| MLP_(4L,16N)_ELU | 0.2673 |
| MLP_(4L,16N)_ISLU[0] | 0.3721 |
| MLP_(4L,16N)_ISLU[1]b | 14799.9303 |

Table 5: Table for Figure 12

| Model | Score |
|---|---|
| WGN_(4L,64N)_ISLU[1]b_MB | 0.0023 |
| WGN_(4L,16N)_ISLU[1]b_MB | 0.0057 |
| WGN_(4L,15N)_ISLU[1]b_MB | 0.0073 |
| WGN_(4L,64N)_ELU_MB | 0.0248 |
| WGN_(4L,64N)_ISLU[1]b_ST | 0.0330 |
| WGN_(4L,16N)_ELU_MB | 0.0472 |
| WGN_(4L,64N)_ELU_ST | 0.0487 |
| WGN_(4L,16N)_ISLU[1]b_ST | 0.2064 |
| WGN_(4L,16N)_ELU_ST | 0.3642 |

Table 6: Table for Figure 13

| Model | Score |
|---|---|
| WGN_(4L,16N)_ISLU[1]b_MB | 0.0057 |
| WGN_(4L,16N)_ISLU[0]_MB | 0.0073 |
| WGN_(4L,16N)_ISLU[1]a_MB | 0.0137 |
| sWGN_(4L,16N)_ISLU[1]b_MB | 0.0161 |
| WGN_(4L,16N)_ELU_MB | 0.0472 |
| MLP_(4L,64N)_ISLU[1]b | 0.0479 |
| MLP_(4L,64N)_ISLU[0] | 0.0650 |
| sWGN_(4L,16N)_ELU_MB | 0.0656 |
| MLP_(4L,64N)_ELU | 0.0949 |
| MLP_(4L,64N)_ISLU[1]a | 0.1086 |
| mMLP_(4L,16N)_ISLU[1]b | 0.2243 |
| MLP_(4L,16N)_ELU | 0.2673 |
| MLP_(4L,16N)_ISLU[0] | 0.3721 |
| mMLP_(4L,16N)_ISLU[0] | 0.3780 |
| MLP_(4L,16N)_ISLU[1]a | 0.3852 |
| mMLP_(4L,16N)_ISLU[1]a | 0.5812 |
| mMLP_(4L,16N)_ELU | 0.8616 |
| MLP_(4L,16N)_ISLU[1]b | 14799.9303 |

Table 7: Table for Figure 14

| Model | Score |
|---|---|
| WGN_(4L,16N)_ISLU[1]a_MB | 0.0005 |
| WGN_(4L,16N)_TANH_MB | 0.0008 |
| WGN_(4L,16N)_ELU_MB | 0.0009 |
| MLP_(4L,64N)_ISLU[1]a | 0.0010 |
| MLP_(4L,64N)_ELU | 0.0013 |
| MLP_(4L,16N)_ISLU[1]a | 0.0021 |
| MLP_(4L,16N)_ELU | 0.0031 |
| MLP_(4L,64N)_TANH | 0.0031 |
| MLP_(4L,16N)_TANH | 0.0052 |

Table 8: Table for Figure 15

| Model | Score |
|---|---|
| WGN_(4L,16N)_ISLU[1]b_MB | 0.0032 |
| MLP_(4L,64N)_TANH | 0.0237 |
| WGN_(4L,16N)_ELU_MB | 0.0272 |
| MLP_(4L,64N)_ELU | 0.0475 |
| MLP_(4L,64N)_ISLU[1]a | 0.1189 |
| WGN_(4L,16N)_TANH_MB | 0.1217 |
| MLP_(4L,16N)_TANH | 0.1290 |
| MLP_(4L,16N)_ELU | 0.1387 |
| MLP_(4L,16N)_ISLU[1]a | 0.1903 |

# F SMOOTHNESS SCORE

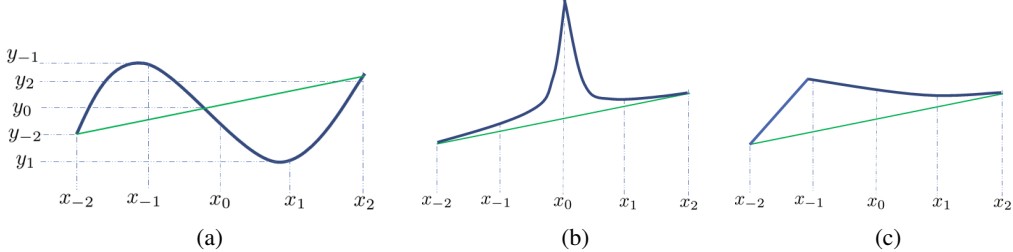

Figure 20: (a) shows the case where the degree of waviness in a given section can be checked. Meanwhile, cases such as (b) and (c) cannot be checked well.

In this study, the smoothness score is defined as a temporary measure, and the smoothness of some tested models is examined.

The smoothness score judges "how wavy." To determine this, *the smoothness of a particular period* is first considered because there is a need for a standard to determine whether the case of Figure 21a or Figure 21b is smooth.

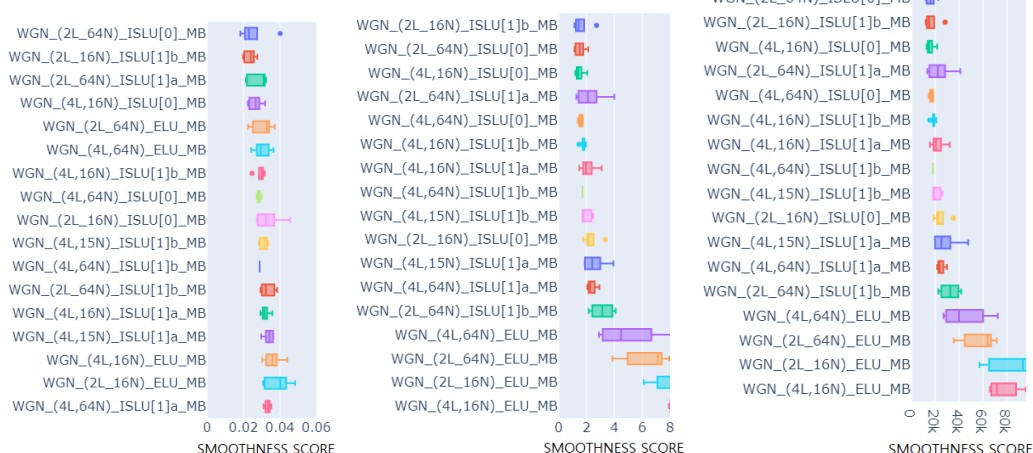

(a) The smoothness score of the 0th derivative

(b) The smoothness score of the 1st derivative

(c) The smoothness score of the 2nd derivative

Figure 22: The smoothness scores with the period 0.007 in all directions for WGN models are shown.

First, the function value $y_{-2}, y_{-1}, y_0, y_1, y_2$ are selected for the uniform interval $x_{-2}, x_{-1}, x_0, x_1, x_2$. Let the path connecting $(x_{-2}, y_{-2})$ and $(x_2, y_2)$ be $y = L(x)$. If the signs of $y_{-1} - L(x_{-1})$ and $y_1 - L(x_1)$ differ, the path is nonsmooth. For A and B, $|A| + |B| - (A + B)$ is 0 if both $A \, and \, B$ have the same sign; otherwise, a positive number greater than 0 is obtained. Therefore, using this, the degree of smoothness of the path is considered as $(|y_{-1} - L(x_{-1})| + |y_1 - L(x_1)| - (y_{-1} - L(x_{-1})) + y_1 - L(x_1)))$, and its total sum is taken as the smoothness score. In a multidimensional case, the sum of smoothness scores in all directions is the total smoothness score.

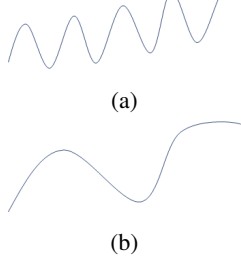

(a)

(b)

Figure 21: Waviness

The disadvantage of this smoothness concept is that it cannot comprise the cases where there is a peak and a sudden bend (Figures 20b and 20c), although such cases can be checked by considering the differential of smoothness or smoothness of a smaller period.

This smoothness score has two parameters: the *period*, which determines the size of the section to be checked, and the moving *step* interval. Figure 22 shows the smoothness for several WGN models with period 0.007 and step 0.007. The figure shows the following two results: (1) ISLU is smoother than ELU and (2) the fewer the parameters and layers, the lower the smoothness score.

