# OpenReview forum: "Smooth Mathematical Functions from Compact Neural Networks"
_ICLR.cc/2023/Conference — Submitted to ICLR 2023_

### Official Review · Reviewer_uCcd · 2022-10-18

**Confidence:** 3
**Correctness:** 2
**Technical Novelty And Significance:** 2
**Empirical Novelty And Significance:** 2
**Recommendation:** 3

**Clarity, Quality, Novelty And Reproducibility:**

Clarity and reproducibility
The comprehensibility of the paper differs from one section to another. Some parts of the paper are written clearly with well-chosen illustrative Figures, while others are not:
- Namely the WGN model and meta-batch concept (Section 3.1, 3.2) are not described comprehensibly and in enough detail. This fact also lowers down reproducibility of the results.
What is the inner structure of the Weight Generator sub-network?  I also don't understand the principle of how exactly the two parts of the network are connected.  Even the Figure 10 does not help much.
Could you describe the model, its parts and their inputs more clearly? Explanation on task <1> might help.
- The training algorithm and the way how the error is propagated through the model should also be explained in more detail (e.g., using a pseudo-code).
- It is not clear how the concepts explained in 2.2 and 2.3 and in Equation (1) are used in the WGN model.

- The Abstract is a bit confusing. It is more common for a scientific publication to place such a section-wise list of content at the end of the Introduction section rather than into the Abstract.

- The paper should be proof-read for language issues (especially Section 1).

Novelty
- The solved task is interesting. The presented WGN model and the methodology as a whole seem to be reasonably novel. However, it is questionable, how useful and significant the contribution is (better experimental evaluation might help to answer this question).
- Some important relative work is not mentioned and analyzed (see "Strength And Weaknesses" for details).

Quality
- The paper suffers for lower technical quality from the experimental point of view.  See "Strength And Weaknesses" section for details.

Questions
- You argue that ISLU activation function leads to smoother and less wavy network function.
To better show the smoothness of ISLU, could you provide a Figure (similar to Figure 3) with also second derivatives of ISLU compared to the other mentioned activation functions (softplus, ELU, hyperbolic tangent,...)? Can you explain your claim that the problem of the hyperbolic tangent function is, that "the output function bends in two places after each layer."?




**Strength And Weaknesses:**

Plus
- The paper looks at accurate regression from an interesting point of view. The authors are looking for "unwavy" network functions (they call them "smooth" which is slightly misleading according to the common sense of the word)
- The concept of meta-parameters of the represented mathematical function that can serve as hints during training or to create meta-batches is interesting and worth further research.
- The WGN-model seems to be novel enough, its combination with with smooth activation function and the presented meta-batch concept to achieve better results looks reasonable.

Minus
1) The experiments are not well chosen to highlight advances and usefulness of the presented model:
- I am not sure, whether the presented model/concept isn't unnecessarily complex to succeed on the presented artificial tasks that are relatively simple.
  I would guess, that similar compact and smooth models could be created in a much simpler way (than WGN)
  by one (or a combination) of the classical MLP-techniques for generalization improvement and/or for reduction of model VC-dimension.
  What about classical learning from hints - the variant where meta-parameters are used as hint-outputs during training seems promising. Also pruning, well-chosen regularization,...
  Further experiments comprising at least learning with hint outputs could rebut such suspicion.
- The experimental evaluation doesn't comprise any real world data. Many models work well for simple artificial tasks but have problems on real world data that is is usually less regular/noise-corrupted/not perfectly distributed etc. Could you enhance the evaluation by such data?
- The only tested criterion is "smoothness score" that measures "waviness" of the network function. The measure is approximate and measures presence of just one type of waves. It might thus unfairly advantage one technique over others.
   If you alter the "smoothness score" to e.g., identify also peaks, it might lead to more fair comparison.
-  There is another argument, why it is not good to use "smoothness score" as the only criterion - best (zero) scores will be achieved by MLPs with only linear or even constant activation functions that however won't be able to learn the task at al.
  - Therefore, why do you omit common measures, such as generalization error or sensitivity (i.e., derivative of the model outputs with respect to its inputs), as further tested criteria?
  If the network function is "too wavy," its vc-dimension is "higher then enough" and it is indicated also by greater generalization error. Smooth network function is also characterized by low sensitivity.

2) The paper omits classical research on the topic - there are several general methods how to form compact, smooth and accurate MLP-models (i.e., regularization, pruning, learning from hints, sensitivity analysis,...).
   A citation and discussion of such techniques would be appropriate. E.g., the concepts presented in 2.2. and 2.3 (or even 3.2) are obvious variants of learning from hints.
3) The core of the paper - Sections 3.1 and Section 3.2 are not written comprehensibly. The details of the model and the training process are not described adequately.



**Summary Of The Paper:**

The authors present a complex methodology of how to generate compact MLP-models that would highly accurately and smoothly represent mathematical functions.
Core of the methodology (contributions):
1. a novel hierarchical MLP-model called WGN (weight generating network) - one part of the model ("weight generator") generates parameters of the later part ("main network").
2. a novel activation function ISLU - a smooth version of ELU, concurrent to Softplus (it promises "less wavy" second derivatives)
3. a meta-batch concept - meta-parameters of the represented mathematical function are used as hints during the training
4. "smoothness score" - an approximate measure that evaluates how "wavy" a network function is

Experimental evaluation comprises three relatively simple artificial datasets. It compares the MLP and WGN models with "smoothness score" as the tested criterion.



**Summary Of The Review:**

The solved task is interesting and the presented model seems to be novel enough.
If the authors provide an adequate experimental evaluation to show usefulness of their approach and if they improve both description of the WGN model and settle their approach better into the related work, I would change my score towards accepting the paper.

---

### Official Review · Reviewer_Mwpn · 2022-10-21

**Confidence:** 3
**Clarity, Quality, Novelty And Reproducibility:** See 'Strength And Weaknesses' section
**Correctness:** 1
**Technical Novelty And Significance:** 1
**Empirical Novelty And Significance:** Not applicable
**Recommendation:** 1

**Strength And Weaknesses:**

Strength
* None

Weakness
* The paper is not well structured nor formally described. For example:
    * What is the formal definition of 'well-developed' in p.2
    * What is the meaning of 'good' in p.3
    * Most figures are not cited from the main manuscript
    * It is very hard to understand the experimental settings
    * It is very hard to understand the definition of 'metadata'


**Summary Of The Paper:**

This paper proposes a new activation function named integrated sigmoid linear unit (ISLU), and evaluates its performance on a regression task.

**Summary Of The Review:**

See 'Strength And Weaknesses' section

---

### Official Review · Reviewer_QykB · 2022-10-22

**Confidence:** 5
**Correctness:** 3
**Technical Novelty And Significance:** 2
**Empirical Novelty And Significance:** 2
**Recommendation:** 3

**Clarity, Quality, Novelty And Reproducibility:**

The clarity of the paper is missing, the English used is weak. However the notations and nomenclatures are fine.
Novelty in the manuscript is missing because the proposed activation function is just a modified ELU with additional parameters for better tuning. Moreover, as meta-parameters are important for improving the performance, the authors have not presented a robust method of computing them.
The method is not reproducible; since the authors have not shared their code via online repositories like github. At least, I can't find anything except the PDF. I'm usually not rigid but it's unlikely that my opinion about the paper will change.


**Strength And Weaknesses:**

An idea is presented by the authors to augment data and meta-parameters for improving the performance of NN. A novel activation function which works very close to ELU is also worth exploring.

Weaknesses
There is no theoretical foundation of the work. When the authors say that the presence of meta parameters improves the performance of NN, is there any theory that validates the claim. If so then they must present it. Otherwise the improved performance could be just a mere coincidence on the data presented in the manuscript. The data set used by the authors is generated from the formula in the physical concept of a spring connected to a support at one end and has a mass m attached at the other end.
Another validation needed in the manuscript is the proof of Universal approximation theorem for the new activation function ISLU. Does ISLU satisfy UAT!
ISLU is a just modified version of ELU with additional parameters added to ELU, alpha and beta, to tune the curvature of the output function at every layer.  If there is more to it, the authors need to mention that. Like how is the AF unique?
Regarding the architecture of NN, it is expected from authors to explore why they have chosen 4 layers for training adding/reducing layers can affect the results! Or Is it because the NN performs fine with 4 layers for that specific dataset used by authors. So the model will work fine if the dataset is changed?
It is important to analyse if the AF, and the concept of meta parameters and weight generating network works well for all types of datasets and on the other regression tasks.


**Summary Of The Paper:**

The authors have worked on NNs that can generate or rather approximate accurate and smooth function mainly to solve problems related to regression. The NN uses a few weight parameters, for regression.  They have reinterpreted the outputs of NNs and have proposed a new activation function–integrated sigmoid linear unit (ISLU).
They have captured the essence of metadata, meta-parameters (called fictitious meta-parameters) for improving the performance of neural networks used in regression.
Some calculations of the form of the activation presented in the paper reveal the following:
log(alpha + exp(beta))/beta  - log(1+alpha)/beta
log[alpha/(alpha+1)+ exp(beta*x)/(1+ alpha)]
log[(alpha(1+alpha))(1 + exp(beta*x)/alpha)]
Some constant + log(1 + exp(beta*x)/alpha) which is a minor variation of the activation function log(1 + exp(beta*x)) which is known to have a saturation problem. I don’t see a major contribution in the existing cottage industry of activations.


**Summary Of The Review:**

The manuscript is a bit weak with respect to empirical as well as theoretical validations. Empirically, testing a model requires division of data into train and validation sets to ensure that there is no bias while experimenting. Moreover, there is just one dataset used during experimentation which is generated via a physical system of a spring attached at one end. Such physical systems (spring-mass) are now repeatedly used to describe activation functions (DiffAct, Saha et al, IJCNN 2021). The solution to the system is a polynomial where coefficients can be learned from data.
The NN regression should be rigorously checked on multiple datasets. The authors have not clarified the method used by the authors to generate hyper-parameters in their manuscript. Theoretical validations of the new activation function are missing as well. The authors need to show if the new AF can approximate any function with a fair degree of accuracy.

---

### Official Review · Reviewer_DP91 · 2022-10-24

**Confidence:** 3
**Correctness:** 1
**Technical Novelty And Significance:** 1
**Empirical Novelty And Significance:** 1
**Recommendation:** 1

**Clarity, Quality, Novelty And Reproducibility:**

I find this paper very confusing. Despite my efforts, I could not understand the problem setup or the significance, and I needed to constantly guess what the authors are trying to say and what was being described. Here is a list of things that are either syntactically incorrect or semantically confusing.

Abstract
“This is paper for” This feels really weird to read, I think this entire sentence doesn’t convery any additional information given the paper title, so this sentence can be safely removed.
“In this study, we get NNs” It’s a bit confusing here, what do you mean by “we get”? do you mean you trained, or you took some pretrained/existing NNs from somewhere?
In the same sentence, “through discussing a few topics about regression”,  you mean you get NNs through discussing a few topics about regression? Sorry that makes no sense to me.
“the one of a simple hierarchical NN that generate models substituting mathematical function is presented”
“the new batch concept “meta-batch" which improves the performance of NN several times more is introduced.” This sentence makes partial sense, but I think it’s much better if authors don’t use the passive voice here: ““we introduce a new batch concept “meta-batch" which improves the performance of NN several times more.”

Introduction
After reading the introduction, I’m still pretty confused about what this paper is about. Key concepts like metaprameters and meta-batch were not explained at all.


NNs for Regression
“They can be seen as basis functions” Do authors mean basis functions in the sense of rigorous mathematics? If so, authors would need to provide theoretical justification or point to prior literature. If not, authors should explain what exactly do they mean by basis functions.
I’m not sure what is the point of Figure 2. I’m not sure what’s the point of Figure 1 either.
“If a one-dimensional regression problem is modeled with a simple MLP that has (k+1) layers with nodes [N0, N1, N2..Nk], the output function will bend more than N0 ∗ N1...Nk.” What do authors mean by “bend more than”?

“Thus, the question is which activation function can develop the intermediate basis functions well? If the activation function starts as a linear function and bends at an appropriate curvature after each layer, the final result will be good.” This seems to be the motivation for the new activation function proposed by the author, but I couldn’t understand this sentence. I understand every word, but I can’t understand what’s going on.

“There is a significant difference in performance between SoftPlus and ISLU.” What difference? Is it desired, or not desired? What’s the hypothesis/prediction of the authors’ theory?



I would advise authors to read Steven Pinker’s The Sense of Style and revise the paper according to the principles described in the book. That’ll be a gift to make the lives of the reviewers and potential readers much better. For example, I noticed that authors excessively used passive voice, passive voice is not always better, and it sometimes makes the writing much harder to read. See the Sense of Style for more details.

At the very least, authors could consider using grammar tools like Grammarly or word tune to check for grammatical mistakes, or proofread before submitting. This is only fair to the reviewer/reader’s time.

“When using metadata, the performance is improved because biases are determined by referring to various data.” What is metadata? It is still not described though it has been referred to many times.

2.1 PERSPECTIVES OF METADATA
Finally metadata is introduced, but it is still not clear to me what is the problem set up and I needed to guess. Do the authors mean that for the regression problem we have access to parameters of the data-generating process? And those parameters are metadata? But if we already have parameters and the data-generating process, it doesn’t seem like a significant problem to begin with, so I remain extremely uncertain of what is going on.

2.3 LEARNING FUNCTION WITH RESTRICTED METADATA
This section also makes no sense to me. The comparison seems to be to be mostly about intrinsic data dimension. If you call representative points on a polynomial metaparameters, you can also call representative images metaparameters? Like eigen faces in a small face image dataset?
3.1 WGN
“We consider the one of the structure of a function-generating network called weight generating network(WGN) in this study.” This sentence makes no sense to me either, and this is the point where I decided to give up … I’m sorry but I don’t think authors have finished writing this paper, and spending any more time on reviewing an unfinished, not readable paper is not justified and not fair for reviewer’s time.



**Strength And Weaknesses:**

Strength: Perhaps after this paper is finished there will be strengths, but currently I can’t find any.

Weakness: See next section


**Summary Of The Paper:**

I don’t think this paper is finished and ready to be reviewed.

**Summary Of The Review:**

I don’t think this paper is finished and ready to be reviewed.

---

### Decision · Program_Chairs · 2023-01-20

**Decision:**

Reject

**Justification For Why Not Higher Score:**

See above.

**Justification For Why Not Lower Score:**

N/A

**Metareview: Summary, Strengths And Weaknesses:**

Reviewers had a variety of concerns about clarity, theoretical motivation, discussion of prior work, and experimental validation. Since the authors have not submitted a response, I will go with the reviewers' judgment and recommend rejection.